# Shifting respiratory pathogens: Post-COVID-19 trends in community-acquired infections in underserved communities

Rayane Rafei[1], Marwan Osman[2]*, Bashir Amer Barake[3], Hassan Mallat[1], Fouad Dabboussi[1], Monzer Hamze[1,4]*

1 Laboratoire Microbiologie Santé et Environnement (LMSE), Doctoral School for Science and Technology, Faculty of Public Health, Lebanese University, Tripoli, Lebanon, 2 Department of Neurosurgery, Yale University School of Medicine, New Haven, Connecticut, United States of America, 3 LAU Gilbert and Rose-Marie Chagoury School of Medicine, Byblos, Lebanon, 4 Clinical Laboratory, Nini Hospital, Microbiology Department, Tripoli, Lebanon

* marwan.osman@yale.edu, mo368@cornell.edu (MO); mhamze@ul.edu.lb (MH)

## Abstract

Respiratory tract infections, caused by various bacteria and viruses, pose a significant global health burden. In Lebanon, post-COVID-19 epidemiological data on respiratory infections remain scarce. To address this gap, this multicenter study investigates the epidemiology of community-acquired acute respiratory infections among children and adults in Tripoli, North Lebanon. From May 2023 to February 2024, nasopharyngeal samples were collected from outpatients with acute respiratory infections visiting hospitals and pediatric clinics in Tripoli. Samples were analyzed using BioFire® Respiratory Panel 2.1 Plus (bioMérieux, France), which targets 23 pathogens, including 19 viruses and four bacteria. We used multivariable logistic regression models to identify the determinants of respiratory infections and examine associations between respiratory pathogens. Among 324 enrolled patients, 69.1% were co-infected with at least one pathogen. Human rhinovirus/enterovirus was the most prevalent (27.2%), followed by influenza A (19.8%), particularly influenza A/H1-2009 (16.4%), and RSV (11.4%). SARS-CoV-2 was still circulating with a prevalence of 6.8%. Classical human coronaviruses accounted for 6.1% of infections, with HCoV-NL63 (2.8%) being the most common. Parainfluenza viruses were identified in 5.2% of patients, with type 4 (2.5%) being the most prevalent, followed by type 3 (1.5%), type 1 (1.2%), and type 2 (0.3%). Logistic regression analysis revealed that human rhinovirus/enterovirus infection decreased the likelihood of influenza A (OR=0.25; 95%CI=0.10–0.54; P=0.001) or SARS-CoV-2 (OR=0.21; 95%CI=0.03–0.75; P=0.039) co-infection. Additionally, our logistic regression models identified significant associations between various determinants, symptoms, and common viruses, including a lower likelihood of influenza A (OR=0.23; 95%CI=0.06–0.76; P=0.019) and RSV (OR=0.29; 95%CI=0.10–0.76; P=0.017) infection among patients with

**Data availability statement:** The raw data and R code required to replicate the analysis are publicly accessible (DOI: 10.5281/zenodo.14866744).

**Funding:** This work was supported by a grant from the Lebanese University, Al Hamidy Medical Charitable Center, the GABRIEL Network, and Mérieux Foundation. The funders had no role in study design, data collection and analysis, decision to publish, or preparation of the manuscript.

**Competing interests:** The authors have declared that no competing interests exist.

higher educational levels. Notably, parainfluenza virus infections occurred significantly more in refugee patients (OR=7.22; 95%CI=1.19–37.0; P=0.020) compared to the host community. In conclusion, this study provides critical insights into the post-pandemic epidemiology of respiratory infections in Lebanon, informing clinicians, health authorities, and policymakers to optimize diagnostics, preventive measures, and antimicrobial stewardship strategies.

## Introduction

Respiratory tract infections continue to be one of the common causes of morbidity and mortality worldwide. In 2021, COVID-19 was the second leading cause of death, following ischemic heart disease, while lower respiratory tract infections (excluding COVID-19) ranked as the fifth leading cause of death [1]. In 2019, there were an estimated 17.2 billion global incident cases of upper respiratory infections, accounting for 42.8% of all cases reported in the Global Burden of Diseases, Injuries, and Risk Factors Study 2019 [2]. While typically short-term, mild, and self-limiting, upper respiratory tract infections can cause severe complications like pneumonia, glomerulonephritis, and myocarditis, creating a substantial burden on both individuals and society [2]. For instance, acute respiratory tract infections result in at least 300 outpatient visits per 1,000 persons per year among children and adolescents in the United States [3].

While bacteria could cause acute respiratory tract infections, most are attributed to viruses [4]. Respiratory viruses have gained particular interest in the past few decades due to the concerning emergence of at least four respiratory pathogens in humans. These pathogens exhibit generally high infectivity and transmissibility overwhelming an immunologically naive population and thus threatening public health: severe acute respiratory syndrome coronavirus (SARS-CoV), influenza A H1N1 pdm09 (2009 H1N1), Middle East respiratory syndrome coronavirus (MERS-CoV), and severe acute respiratory syndrome coronavirus 2 (SARS-CoV-2) [5–7]. Since its first identification in Wuhan, China, in December 2019, SARS-CoV-2, the causative agent of Coronavirus Disease (COVID-19), has become one of the global health emergencies in recent history, leading to more than 776 million reported COVID-19 cases as of August 18, 2024. Lebanon reported its first confirmed COVID-19 case on February 21, 2020. According to the World Health Organization (WHO), as of December 19, 2023, the country has recorded over 1.2 million confirmed cases and 10,947 deaths attributed to the virus [8]. Earlier in the pandemic, the Lebanese government implemented several strict public and social measures and successfully delayed the steep spike in COVID-19 cases and hospitalizations [9]. However, the cumulative effect of numerous factors, including the severe economic crisis, the explosion of the Beirut port, the lifting and easing of implemented measures, and the appearance of new SARS-CoV-2 variants brought about four waves since the first confirmed case to the end of June 2022. These waves are characterized by a sustained increase followed by a decline in daily COVID-19 cases [9]. Indeed, the severe Lebanese

economic and financial crisis has weakened the healthcare system and caused a drastic shortage of medical devices and essential medications, exacerbating the control of the COVID-19 epidemic in Lebanon. The massive Beirut blast on August 4, 2020 has further strained the already fragile healthcare system, resulting in 220 deaths, 6,500 people injured, and 300,000 displaced, impacting half of Beirut's healthcare centers and leading, among other causes, to a substantial increase in COVID-19 cases in wave 1. The emergence of Alpha, Delta, and Omicron variants has also contributed in part to the witnessed Lebanese COVID-19 waves, namely waves 2 (February 2021-June 2021), 3 (July 2021-October 2021), and 4 (December 2021-June 2022), respectively [9,10]. Deployment of COVID-19 vaccines launched on February 14, 2021, succeeded in covering 50.3% of the Lebanese population for the first dose as of December 12, 2022, and decreased the hospitalization rates during the third and fourth waves of the pandemic compared to the second wave [9].

The clinical diagnosis of respiratory infections is challenging owing to many co-circulating respiratory pathogens and the overlap of signs and symptoms between causative pathogens, resulting thus in very poor specificity of symptomatic surveillance [11,12]. Co-infections may also complicate diagnosis and increase the risk of complications and disease severity [13]. A timely and accurate identification of the etiological agent responsible for a respiratory infection is quintessential for clinical management, including prescribing appropriate therapy, decreasing antimicrobial therapy, predicting disease severity, improving patient prognosis, and reducing economic burden [12,14–16]. Diagnosing respiratory infections has undergone a quantum leap in the molecular era with the advent of multiplex assays. These advanced tools enable rapid, comprehensive screening of multiple pathogens from respiratory samples, facilitating the detection of co-infections, monitoring pathogen circulation, and anticipating emerging threats. By filling critical knowledge gaps, these assays enhance our understanding of respiratory disease epidemiology and improve outbreak preparedness by enabling early-phase detection and intervention [17–19].

The emergence of COVID-19 has impacted not only humans but also other respiratory pathogens. Indeed, public health measures enacted during the pandemic have caused changes in seasonality, disappearance, or significant reductions over extended periods in different parts of the world in respiratory virus infections other than COVID-19 [20–23]. Despite being scant and generally conducted in Beirut, data about the type of respiratory viral and bacterial infections before COVID-19 existed in Lebanon [24–27]. However, data are nearly absent during and after COVID-19. Our overarching goal is, therefore, to investigate the epidemiology of community-acquired acute respiratory infections in the post-COVID-19 (2023−2024, after 4 years of COVID-19 onset) among children and adults in Tripoli, North Lebanon using a multiplex assay BioFire® FilmArray® Respiratory Panel 2.1 Plus. Specifically, we sought to (i) unravel the prevalence of screened respiratory pathogens, (ii) identify the predominant pathogens responsible for respiratory infections among outpatients and children visiting pediatric clinics, (ii) determine risk factors and possible disease temporality, (iv) unravel potential associations between pathogens among patients with acute respiratory symptoms. As our study was done after COVID-19 onset, we will try to assess any potential change in the landscape of respiratory pathogens after COVID-19 by comparing our data to Lebanese pre-COVID-19 studies.

## Materials and methods

### Ethics statement

This investigation was conducted according to the Declaration of Helsinki and approved by the institutional review board (IRB) of the Azm Center for Research in Biotechnology & its Applications, Doctoral School of Science and Technology at the Lebanese University (CE-EDST-3–2023, approval notice: January 23, 2023). Written informed consent was obtained from all individuals involved in the study.

### Study design

This multicenter cross-sectional community-based study was conducted at the Al Hamidy Medical Charitable Center, Nini Hospital, and four private primary care pediatric clinics located throughout Tripoli, the capital city of North

Governorate. The population eligible for the study included patients of all ages who were clinically diagnosed with acute community-acquired respiratory infections and attended one of the healthcare facilities between May 2023 and February 2024. Inpatients and patients receiving antibiotics were excluded from the study. After completing an informed consent, the patient was included in this study, and a nasopharyngeal swab was collected using a sterile flocked swab. Due to the academic recess at Lebanese University, sample collection was paused at the end of July and resumed in September.

Participants or their legal representatives filled out a questionnaire covering sociodemographic characteristics (e.g., age, sex, residence, nationality, social status, educational level), respiratory and gastrointestinal symptoms (e.g., fever, runny nose, headache, cough, wheezing, diarrhea), and their vaccination status for influenza and COVID-19. To achieve the study objectives with a 5% margin of error and a 95% confidence level, the minimum required sample size was calculated as 246 using the Raosoft sample size calculator (http://www.raosoft.com/samplesize.html) based on a population size of 1,000,000 and an estimated respiratory pathogen prevalence of 80% [19].

## Microbiological analysis of biological samples

After collection, nasopharyngeal swabs were immersed in saline medium and immediately transported in sterile containers at 4°C to the Laboratoire Microbiologie Santé et Environnement (LMSE) in Tripoli, Lebanon, where they were promptly processed. All samples were analyzed using the BioFire® FilmArray® Respiratory Panel 2.1 Plus (bioMérieux, France), a fully automated multiplex PCR system, following the manufacturer's instructions and using a fixed volume of the sample storage medium. No additional microbiological testing (e.g., culture) was performed. This panel could identify 23 pathogens: 19 viral targets (adenovirus, human coronavirus OC43 [HCoV-OC43], human coronavirus HKU1 [HCoV-HKU1], human coronavirus 229E [HCoV-229E], human coronavirus NL63 [HCoV-NL63], human metapneumovirus, human rhinovirus/enterovirus, influenza A virus, influenza A virus A/H1, influenza A virus A/H3, influenza A virus A/H1-2009, influenza B virus, parainfluenza virus 1 [PIV1], parainfluenza virus 2 [PIV2], parainfluenza virus 3 [PIV3], parainfluenza virus 4 [PIV4], the respiratory syncytial virus [RSV], Middle East respiratory syndrome coronavirus [MERS-CoV], severe acute respiratory syndrome coronavirus 2 [SARS-CoV-2]) and 4 bacterial targets (*Bordetella parapertussis*, *Bordetella pertussis*, *Chlamydia pneumoniae*, and *Mycoplasma pneumoniae*). The clinicians and their patients were swiftly informed of positive and negative results.

## Statistical analysis

The data collected from the questionnaire and laboratory results were reviewed for completeness and consistency prior to analysis. Statistical analysis was performed using R software (R Core Team, version 4.4.0; R Studio, version 2024.04.2–764). The dataset was imported for data cleaning, variable coding, and analysis. Descriptive statistics were applied to all variables and visualizations were created using the ggplot2 package. All raw data and R code required to replicate the analysis are publicly accessible (https://doi.org/10.5281/zenodo.14866744). Continuous variables were presented as mean ± standard deviation [min-max], while categorical variables were shown as frequency distributions. To identify determinants of respiratory infections at the bivariate level, group differences were assessed using the Pearson chi-squared test for categorical variables. We predicted the determinants of respiratory infections using bivariate and multivariable logistic regression analysis as previously described [28]. Specifically, multivariable logistic regression models were used to assess the determinants of (i) respiratory infections and (ii) each pathogen compared to none, with sociodemographic, clinical, and potential risk factors as explanatory variables. Additionally, the associations between respiratory pathogens were examined using a third set of multivariable logistic regression models, where infection with specific respiratory pathogens (prevalence >5%) served as the explanatory variable. Backward elimination was used to identify the most significant predictors of respiratory infections and pathogen associations. All statistical tests were two-sided, with a type I error of $\alpha = 0.05$.

 

### Inclusivity in global research

Additional information regarding the ethical, cultural, and scientific considerations specific to inclusivity in global research is included in the S1 Checklist.

## Results

This study enrolled 324 patients with respiratory symptoms, comprising 196 females and 128 males. The ages of the patients ranged between 1 and 94 years with a mean of $24.2 \pm 21.1$ years old (Table 1). More than 50% of the patients were adults, 30.6% were preschoolers (≤5 years) and 14.3% were children (6–17 years). Most patients were from urban (84.3%) rather than rural regions (15.7%), lived in regular (71.1%) rather than overcrowded environments, and did not receive the influenza vaccine (93.2%). Patients were principally recruited in 2023 (304 patients) with an additional 20 patients enrolled during the first two months of 2024. Most patients were admitted during fall (63.3%) followed by winter (24.1%), summer (6.5%), and spring (6.2%). The mean number of participating patients was $10 \pm 2.8$ per month with the maximum number being in December, October, and November and the minimum number being in September, January, May, and June. No sample was collected in August (Fig 1).

A total of 224 samples yielded positive results for at least one respiratory pathogen using the BioFire® Respiratory Panel 2.1 Plus (Table 1). Patients were mainly infected by human rhinovirus/enterovirus (27.2%), influenza A viruses (19.8%) particularly influenza A/H1-2009 (16.4%), RSV (11.4%), SARS-CoV-2 (6.8%), classical coronaviruses (6.1%), and PIV (5.2%). No *B. parapertussis*, *C. pneumoniae*, MERS-CoV, and human metapneumovirus cases were detected (Fig 2).

Most positive samples (83%, 187/224) identified one single pathogen, with only 14.7% and 1.8% containing two or three pathogens, respectively (Fig 3A). Single infections consisted mainly of human rhinovirus/enterovirus (34.8%, 65/187), influenza A (24.6%), RSV (12.3%), SARS-CoV-2 (10.2%), PIV (4.8%), HCoV-NL63 (3.7%), HCoV-229E (2.1%), HCoV-OC43 (2.1%), adenovirus (1.6%), influenza B viruses (1.6%), HCoV-HKU1 (1.1%), and *M. pneumoniae* (1.1%) (Fig 3B). Combined infections with two pathogens occurred in 33 patients with human rhinovirus/enterovirus-RSV (18.2%, 6/33), human rhinovirus/enterovirus-influenza A (15.2%), influenza A-RSV (9.1%), human rhinovirus/enterovirus-PIV (9.1%) being the most common combinations (Fig 3B). Four cases had mixed infections with three pathogens: human rhinovirus/enterovirus-influenza A-PIV, human rhinovirus/enterovirus-influenza A-RSV, human rhinovirus/enterovirus-RSV-PIV, and human rhinovirus/enterovirus-RSV-SARS-CoV-2 (Fig 3B).

Temporal trends in the percentage of infection by respiratory pathogens were investigated (Fig 4). Regarding identified bacterial pathogens, *M. pneumoniae* was detected in winter, whereas *B. pertussis* was in spring and summer (Fig 4A). As for common viral pathogens, influenza A virus (including influenza A/H1-2009), RSV, and PIV showed the highest surges in December. Human rhinovirus/enterovirus showed two notable infection peaks, with the October peak displaying a slightly higher percentage than the one in December. Interestingly, SARS-CoV-2 saw one high point in October (Fig 4B).

When assessing a potential link between the sociodemographic factors and acute respiratory infections using the logistic regression model, only a high education level significantly reduced acute community-acquired respiratory infections ($P\text{-value} < 0.001$) (Table 2, S1 Table). Patients suffered from many respiratory and digestive symptoms with vomiting (89%), cough (73.5%), dyspnea (62%), runny nose (57.1%), and fever (50.8%) being the most common (Table 1). However, the backward logistic regression model showed that runny nose and wheezing were significantly observed in infected patients than in non-infected patients ($P < 0.05$) (Table 2, S1 Table). Chest pain and sinusitis were more observed in non-infected than infected patients ($P < 0.05$).

The infection rate by common respiratory pathogens was generally higher in preschool-aged children (≤5 years) compared to adults (Table 3): human rhinovirus/enterovirus (37.5% vs. 21.4%), influenza A virus (28.1% vs. 13.3%), RSV (28.1% vs. 4.1%), PIV (9.4% vs. 3.5%) but not in SARS-CoV-2 (4.2% vs. 9.8%). However, these observations were not statistically significant when running the logistic regression model (Table 4, S2 Table).

**Table 1. Sociodemographic characteristics of the study population and the prevalence of respiratory pathogens.**

| | Total | | Missing information | |
|---|---|---|---|---|
| | N | % | n | % |
| **Age (mean [SD; min-max])** | 24.2 [21.2; 1–94 years] | | 10 | 3.1 |
| **Age class** | | | 10 | 3.1 |
| ≤5 years | 96 | 30.6 | | |
| 6–17 years | 45 | 14.3 | | |
| ≥18 years | 173 | 55.1 | | |
| **Sex** | | | 0 | 0 |
| Female | 196 | 60.5 | | |
| Male | 128 | 39.5 | | |
| **Region** | | | 24 | 7.4 |
| Urban | 253 | 84.3 | | |
| Rural | 47 | 15.7 | | |
| **Educational level** | | | 31 | 9.6 |
| High School and Above | 146 | 49.9 | | |
| Less than High School | 147 | 50.1 | | |
| **Marital status** | | | 24 | 7.4 |
| Married | 97 | 32.3 | | |
| Single | 203 | 67.7 | | |
| **Environment** | | | 0 | 0 |
| Overcrowded | 94 | 29.0 | | |
| Regular | 230 | 71.0 | | |
| **Vaccinated against influenza** | | | 0 | 0 |
| Yes | 22 | 6.8 | | |
| No | 302 | 93.2 | | |
| **Citizenship** | | | 16 | 4.9 |
| Lebanese | 293 | 95.1 | | |
| Syrian or Palestinian refugee | 15 | 4.9 | | |
| **Season** | | | 0 | 0 |
| Summer | 21 | 6.5 | | |
| Fall | 205 | 63.3 | | |
| Winter | 78 | 24.1 | | |
| Spring | 20 | 6.2 | | |
| **Detection of respiratory-tract pathogens** | | | 0 | 0 |
| Yes | 224 | 69.1 | | |
| No | 100 | 30.9 | | |
| *Bacteria* | | | 0 | 0 |
| *Bordetella pertussis* | 4 | 1.2 | | |
| *Bordetella parapertussis* | 0 | 0 | | |
| *Chlamydia pneumoniae* | 0 | 0 | | |
| *Mycoplasma pneumoniae* | 2 | 0.6 | | |
| *Virus* | | | 0 | 0 |
| Adenovirus | 6 | 1.9 | | |
| Coronavirus 229E | 5 | 1.5 | | |
| Coronavirus HKU1 | 3 | 0.3 | | |
| Coronavirus NL63 | 9 | 2.8 | | |
| Coronavirus OC43 | 5 | 1.5 | | |

*(Continued)*

**Table 1.** (Continued)

| | Total | | Missing information | |
|---|---|---|---|---|
| | N | % | n | % |
| MERS-CoV | 0 | 0 | | |
| SARS-CoV-2 | 22 | 6.8 | | |
| Human rhinovirus/enterovirus | 88 | 27.2 | | |
| Human metapneumovirus | 0 | 0 | | |
| Influenza A | 64 | 19.8 | | |
| Influenza A virus (H1-2009) | 53 | 16.4 | | |
| Influenza A virus (H3) | 6 | 1.9 | | |
| Influenza B | 3 | 0.9 | | |
| Parainfluenza virus | 17 | 5.2 | | |
| Parainfluenza virus 1 | 4 | 1.2 | | |
| Parainfluenza virus 2 | 1 | 0.3 | | |
| Parainfluenza virus 3 | 5 | 1.5 | | |
| Parainfluenza virus 4 | 8 | 2.5 | | |
| Respiratory syncytial virus | 37 | 11.4 | | |
| **Respiratory and digestive symptoms** | | | | |
| Vomiting | 275 | 89.0 | 15 | 4.6 |
| Cough | 227 | 73.5 | 15 | 4.6 |
| Dyspnea | 191 | 62.0 | 16 | 4.9 |
| Runny nose | 176 | 57.1 | 16 | 4.9 |
| Fever | 157 | 50.8 | 15 | 4.6 |
| Sore throat | 140 | 45.6 | 17 | 5.2 |
| Malaise | 127 | 41.1 | 15 | 4.6 |
| Headache | 112 | 36.4 | 16 | 4.9 |
| Muscle pain | 111 | 35.9 | 15 | 4.6 |
| Chest pain | 72 | 23.4 | 16 | 4.9 |
| Itchy nose or throat | 59 | 19.2 | 16 | 4.9 |
| Nausea | 44 | 14.2 | 15 | 4.6 |
| Wheezing | 34 | 11.0 | 16 | 4.9 |
| Moderate-to-severe diarrhea | 32 | 10.9 | 30 | 9.3 |
| Sinusitis | 29 | 9.4 | 17 | 5.2 |
| Asthma | 24 | 7.8 | 17 | 5.2 |
| Hypertension | 24 | 7.8 | 17 | 5.2 |

Missing information has been omitted. Data are presented as mean (standard deviation [min-max]) for the continuous variable age and as frequency and percentage for categorical variables.

Multivariable logistic regression models identified that human rhinovirus/enterovirus infections increased in patients with a runny nose (OR = 4.16; 95% CI = 2.19–8.28; P < 0.001) and decreased in those with chest pain (OR = 0.43; 95% CI = 0.19–0.91; P = 0.034) (Table 4, S2 Table). Influenza A virus infections occurred significantly in patients with fever (OR = 7.63; 95% CI = 3.19–20.9; P < 0.001), cough (OR = 3.33; 95% CI = 1.18–11.2; P = 0.034) and vomiting (OR = 4.06; 95% CI = 1.39–12.4; P = 0.011), but decreased in patients with high educational level (OR = 0.23; 95% CI = 0.06–0.76; P = 0.019). Infections by the RSV were significantly lower among patients with a high education level (OR = 0.29; 95% CI = 0.10–0.76; P = 0.017), and suffering from muscle pain (OR = 0.08; 95% CI = 0.0–0.42; P = 0.017) and runny nose (OR = 0.37; 95% CI = 0.15–0.90; P = 0.031), but higher among those with dyspnea (OR = 2.69; 95% CI = 1.11–6.66;

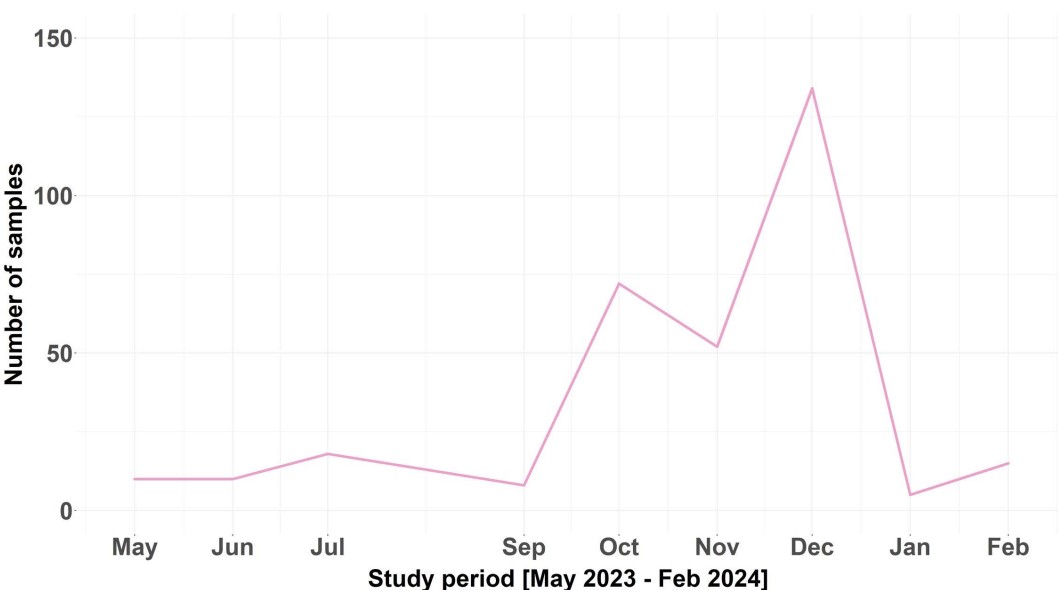

**Fig 1. Total number of collected nasopharyngeal swab samples during the study period (no samples were collected in August 2023).**

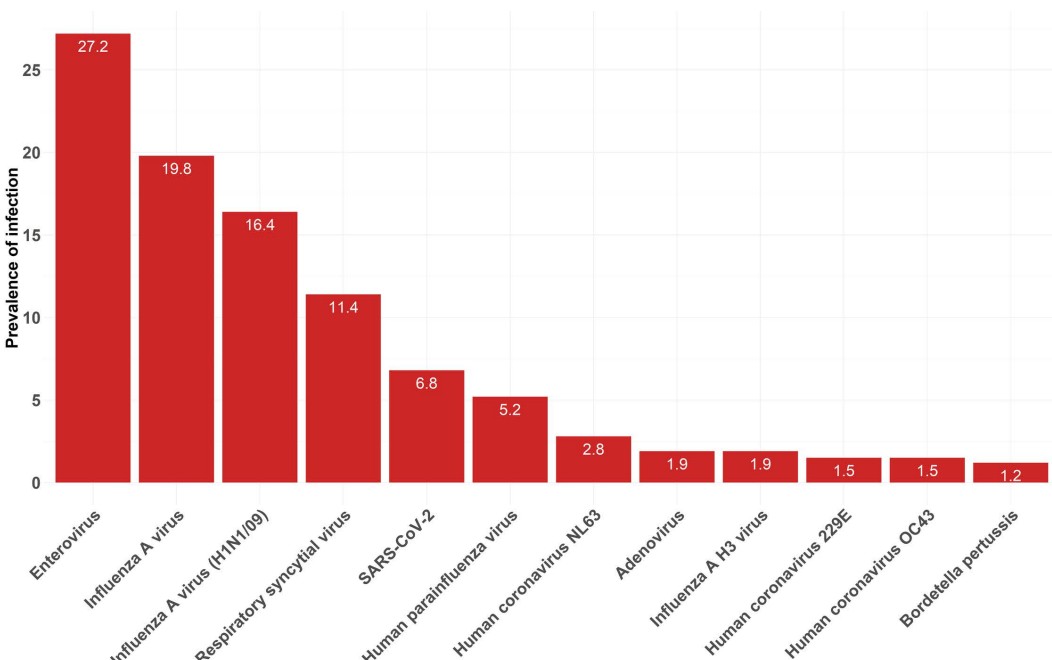

**Fig 2. Prevalence of pathogens among patients suffering from acute community-acquired upper respiratory infections in North Lebanon.**
Enteric pathogens with a prevalence lower than 1% were not shown in this figure (Influenza B virus: 0.9%, *Mycoplasma pneumoniae*: 0.6%, and Coronavirus HKU1: 0.3%). *Enterovirus: Human rhinovirus/enterovirus. The infection rates of influenza A/H1-2009 and influenza A H3 viruses are presented separately as distinct groups; however, both are included in the overall percentage (19.8%) reported for influenza A.

**Fig 3. Distribution of single and mixed infections (A) and most common pathogen association patterns (B) among patients suffering from acute community-acquired upper respiratory infections in Lebanon.** *RSV: Respiratory syncytial virus; PIV: Parainfluenza virus; Enterovirus: Human rhinovirus/enterovirus; SARS-CoV-2: Severe acute respiratory syndrome coronavirus 2; HCoV-229E: Coronavirus 229E; CoV-HKU1: Coronavirus HKU1; CoV-NL63: Coronavirus NL63; HCoV-OC43: Coronavirus OC43.

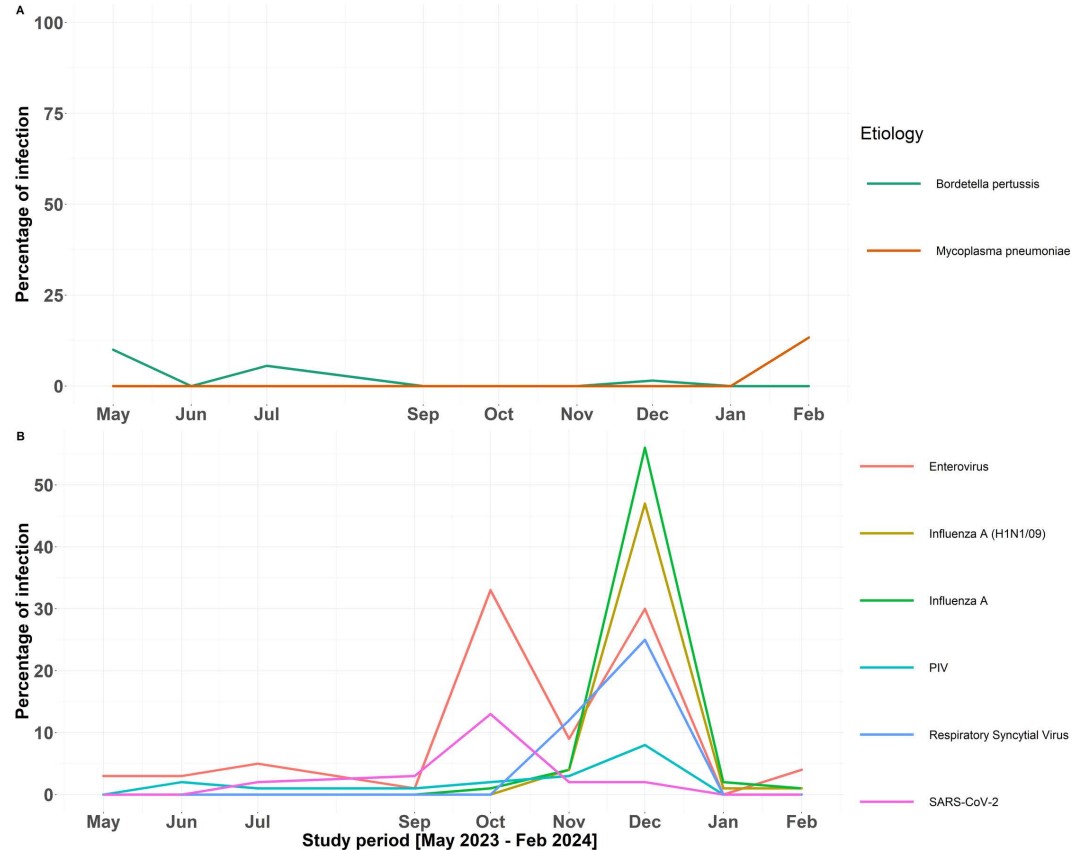

**Fig 4. Temporal trends in the percentage of isolates of pathogens among patients suffering from acute community-acquired upper respiratory infections during the study period. (A)** Bacteria and **(B)** Viruses. (no samples were collected in August 2023). *Enterovirus, Human rhinovirus/entero-virus; PIV, Parainfluenza virus.

P = 0.029). Patients with wheezing were significantly more likely to be infected by SARS-CoV-2 (OR = 6.22; 1.58–24.8; P = 0.008) in contrast to those with chest pain with significantly little likelihood of SARS-CoV-2 infections (OR = 0.12; 95% CI = 0.02–0.56; P = 0.018). Syrian or Palestinian patients experienced substantially higher PIV infections than Lebanese patients (OR = 7.22; 95% CI = 1.19–37.0; P = 0.020) (Table 4, S2 Table).

Multiple logistic regression analysis revealed fewer significant associations between common viruses (with prevalence > 4%) among patients suffering from acute community-acquired respiratory infections in North Lebanon (Table 5, S3 Table). For instance, human rhinovirus/enterovirus significantly reduced the risk of contracting influenza A virus (OR = 0.25; 95% CI = 0.10–0.54; P = 0.001), and SARS-CoV-2 (OR = 0.21; 95% CI = 0.03–0.75; P = 0.039). Influenza A virus infection decreased the probability of being infected by human rhinovirus/enterovirus (OR = 0.24; 95% CI = 0.10–0.53; P = 0.001), whereas SARS-CoV-2 infection lessened the infection likelihood by human rhinovirus/enterovirus (OR = 0.20; 95% CI = 0.03–0.71; P = 0.033).

## Discussion

Our study highlighted the etiology of acute respiratory infections in the fifth year of the COVID-19 pandemic among outpatients and children visiting pediatrician clinics in North Lebanon. This study demonstrates that out of 23 screened pathogens, at least 18 circulated among people with acute respiratory tract symptoms. Although the heterogeneities of adopted

**Table 2.  Sociodemographic and risk factor determinants of acute community-acquired upper respiratory infections in the study population.**

| | Univariate analysis | | Backward multivariable logistic regression model | | |
| --- | --- | --- | --- | --- | --- |
| | Infection | | | | |
| | % | P | adj. OR | 95%CI | P |
| **Age class** | | | | | |
| ≤5 years | **89.6** | **<0.001** | | | |
| 6–17 years | **62.2** | | | | |
| ≥18 years | **59.0** | | | | |
| **Sex** | | | | | |
| Female | 68.9 | 1.00 | | | |
| Male[1] | 69.5 | | | | |
| **Region** | | | | | |
| Urban | 69.2 | 0.396 | | | |
| Rural | 76.6 | | | | |
| **Educational level** | | | | | |
| High school or above | **59.8** | **<0.001** | **0.25** | **0.11-0.53** | **<0.001** |
| Less than High School[1] | **81.4** | | | | |
| **Marital status** | | | | | |
| Single | **76.4** | **0.003** | | | |
| Married[1] | **58.8** | | | | |
| **Environment** | | | | | |
| Overcrowded | 73.4 | 0.352 | | | |
| Regular[1] | 67.4 | | | | |
| **Vaccinated against influenza** | | | | | |
| Yes | 63.6 | 0.734 | | | |
| No[1] | 69.5 | | | | |
| **Citizenship** | | | | | |
| Lebanese | 68.9 | 0.242 | | | |
| Syrian or Palestinian refugee[1] | 86.7 | | | | |
| **Season** | | | | | |
| Fall[1] | **73.3** | **0.002** | | | |
| Other seasons | **53.0** | | 0.48 | 0.20-1.17 | 0.107 |
| **Cough** | | | | | |
| Yes | **73.6** | **0.017** | | | |
| No[1] | **58.5** | | | | |
| **Fever** | | | | | |
| Yes | **74.5** | 0.073 | 1.86 | 0.93-3.77 | 0.081 |
| No[1] | **64.5** | | | | |
| **Nausea** | | | | | |
| Yes | 72.7 | 0.754 | | | |
| No[1] | 69.1 | | | | |
| **Vomiting** | | | | | |
| Yes | 76.5 | 0.467 | | | |
| No[1] | 68.7 | | | | |
| **Dyspnea** | | | | | |
| Yes | 75.2 | 0.114 | 1.81 | 0.84-4.06 | 0.138 |
| No[1] | 66.0 | | | | |
| **Headache** | | | | | |

*(Continued)*

**Table 2.** (Continued)

| | Univariate analysis | | Backward multivariable logistic regression model | | |
|---|---|---|---|---|---|
| | Infection | | | | |
| | % | P | adj. OR | 95%CI | P |
| Yes | 63.4 | 0.104 | | | |
| No[1] | 73.0 | | | | |
| **Malaise** | | | | | |
| Yes | 68.5 | 0.828 | | | |
| No[1] | 70.3 | | | | |
| **Moderate-to-severe diarrhea** | | | | | |
| Yes | 59.4 | 0.180 | 0.43 | 0.15-1.25 | 0.117 |
| No[1] | 72.5 | | | | |
| **Itchy nose or throat** | | | | | |
| Yes | 74.6 | 0.465 | | | |
| No[1] | 68.7 | | | | |
| **Muscle pain** | | | | | |
| Yes | **57.7** | **0.001** | | | |
| No[1] | **76.3** | | | | |
| **Runny nose** | | | | | |
| Yes | **75.6** | **0.011** | **3.98** | **1.99-8.30** | **0.001** |
| No[1] | **61.4** | | | | |
| **Wheezing** | | | | | |
| Yes | **91.2** | **0.007** | **7.57** | **1.80-54.0** | **0.015** |
| No[1] | **66.8** | | | | |
| **Chest pain** | | | | | |
| Yes | **58.3** | **0.028** | **0.35** | **0.14-0.81** | **0.015** |
| No[1] | **72.9** | | | | |
| **Asthma** | | | | | |
| Yes | 54.2 | 0.115 | | | |
| No[1] | 71.7 | | | | |
| **Hypertension** | | | | | |
| Yes | 54.2 | 0.115 | | | |
| No[1] | 71.7 | | | | |
| **Sinusitis** | | | | | |
| Yes | **37.9** | **<0.001** | **0.21** | **0.07-0.59** | **0.004** |
| No[1] | **73.7** | | | | |
| **Sore throat** | | | | | |
| Yes | 70.0 | 0.927 | | | |
| No[1] | 68.9 | | | | |

\* Determinants of infection were predicted using univariate and multivariable analysis. Variables with a P-value ≤ 0.20 in the univariate analysis were included in Model 1 (multivariable logistic regression analysis, Supplementary S1 Table). To further refine the model, a backward logistic regression model was created including only complete cases. [1]Reference group. Bold values indicate statistically significant results.

PCR, targeted pathogens, and settings hinder an effective comparison between studies, the prevalence of positive samples in this study was 69.1%, consistent with other studies reporting high percentages in Lebanon (70%−84%) [24,26] and elsewhere [3]. While the testing panel is extensive, it is not exhaustive. This prevalence could be higher as the screening

**Table 3. Sociodemographic and risk factor determinants of common pathogens (Prevalence > 4%) in acute community-acquired upper respiratory infections using univariate analysis in Lebanon.**

| | Human rhinovirus/ enterovirus | | Influenza A virus | | Respiratory syncytial virus | | SARS-CoV-2 | | Parainflu-enza virus | |
|---|---|---|---|---|---|---|---|---|---|---|
| | % | P | % | P | % | P | % | P | % | P |
| **Age** | | | | | | | | | | |
| ≤5 years | **37.5** | **0.017** | **28.1** | **0.009** | **28.1** | **<0.001** | 4.2 | 0.087 | 9.4 | 0.116 |
| 6–17 years | **28.9** | | **24.4** | | **6.7** | | 2.2 | | 4.4 | |
| ≥18 years | **21.4** | | **13.3** | | **4.1** | | 9.8 | | 3.5 | |
| **Sex** | | | | | | | | | | |
| Female | 27.0 | 1.00 | 18.9 | 0.729 | 10.7 | 0.753 | 7.1 | 0.931 | 5.1 | 1.00 |
| Male | 27.3 | | 21.1 | | 12.5 | | 6.3 | | 5.5 | |
| **Region** | | | | | | | | | | |
| Urban | 25.7 | 0.110 | 20.6 | 0.982 | 10.7 | 0.557 | 7.5 | 1.00 | 5.9 | 0.477 |
| Rural | 38.3 | | 19.1 | | 14.9 | | 6.4 | | 2.1 | |
| **Educational level** | | | | | | | | | | |
| High school or above | 22.6 | 0.101 | **13.4** | **0.003** | **3.7** | **<0.001** | 10.4 | 0.062 | 3.7 | 0.203 |
| Less than High School | 31.8 | | **27.9** | | **20.2** | | 3.9 | | 7.8 | |
| **Marital status** | | | | | | | | | | |
| Single | 30.0 | 0.414 | 23.2 | 0.166 | **15.8** | **0.007** | 6.4 | 0.987 | 7.4 | 0.110 |
| Married | 24.7 | | 15.5 | | **4.1** | | 7.2 | | 2.1 | |
| **Environment** | | | | | | | | | | |
| Overcrowded | 25.5 | 0.983 | 19.1 | 1.00 | 11.7 | 1.00 | 6.4 | 1.00 | 5.3 | 0.813 |
| Regular | 27.8 | | 20.0 | | 11.3 | | 7.0 | | 5.2 | |
| **Vaccinated against influenza** | | | | | | | | | | |
| Yes | 22.7 | 0.813 | 4.5 | 0.115 | 9.1 | 0.993 | 9.1 | 0.996 | 4.5 | 1.00 |
| No | 27.5 | | 20.9 | | 11.6 | | 6.6 | | 5.3 | |
| **Citizenship** | | | | | | | | | | |
| Lebanese | 27.0 | 0.808 | **18.4** | **0.003** | 11.9 | 1.00 | 7.5 | 0.557 | **4.4** | **0.040** |
| Syrian or Palestinian refugee | 33.3 | | **53.3** | | 13.3 | | 0.0 | | **20.0** | |
| **Season** | | | | | | | | | | |
| Fall | 27.9 | 0.658 | **23.6** | **<0.001** | **14.3** | **0.002** | 6.6 | 0.992 | 5.0 | 0.982 |
| Other seasons | 24.2 | | **4.5** | | **0.0** | | 7.6 | | 6.1 | |
| **Cough** | | | | | | | | | | |
| Yes | 27.3 | 1.00 | **26.8** | **0.001** | 14.1 | 0.087 | 5.7 | 0.182 | 6.2 | 0.310 |
| No | 28.0 | | **7.3** | | 6.1 | | 11.0 | | 2.4 | |
| **Fever** | | | | | | | | | | |
| Yes | **21.7** | **0.027** | **31.2** | **<0.001** | 12.1 | 1.00 | 7.6 | 0.887 | 7.0 | 0.224 |
| No | **33.6** | | **8.6** | | 11.8 | | 6.6 | | 3.3 | |
| **Nausea** | | | | | | | | | | |
| Yes | 22.7 | 0.559 | 31.8 | 0.058 | 4.5 | 0.165 | 6.8 | 1.00 | 6.8 | 0.871 |
| No | 28.3 | | 18.1 | | 13.2 | | 7.2 | | 4.9 | |
| **Vomiting** | | | | | | | | | | |
| Yes | 32.4 | 0.641 | **38.2** | **0.010** | 2.9 | 0.150 | 5.9 | 1.00 | 5.9 | 1.00 |
| No | 26.9 | | **17.8** | | 13.1 | | 7.3 | | 5.1 | |
| **Dyspnea** | | | | | | | | | | |
| Yes | 31.6 | 0.226 | 18.8 | 0.758 | **17.9** | **0.020** | 6.0 | 0.696 | 4.3 | 0.760 |
| No | 24.6 | | 20.9 | | **8.4** | | 7.9 | | 5.8 | |

*(Continued)*

| | Human rhinovirus/ enterovirus | | Influenza A virus | | Respiratory syncytial virus | | SARS- CoV-2 | | Parainflu- enza virus | |
|---|---|---|---|---|---|---|---|---|---|---|
| | % | P | % | P | % | P | % | P | % | P |
| **Headache** | | | | | | | | | | |
| Yes | 21.4 | 0.108 | 20.5 | 1.00 | 2.7 | **<0.001** | 8.0 | 0.818 | 3.6 | 0.865 |
| No | 30.6 | | 19.9 | | 17.3 | | 6.6 | | 5.1 | |
| **Malaise** | | | | | | | | | | |
| Yes | 27.6 | 1.00 | 18.1 | 0.567 | **6.3** | **0.017** | 6.3 | 0.807 | 5.5 | 1.00 |
| No | 27.5 | | 21.4 | | **15.9** | | 7.7 | | 4.9 | |
| **Moderate-to-severe diarrhea** | | | | | | | | | | |
| Yes | 12.5 | 0.070 | 28.1 | 0.421 | 6.3 | 0.389 | 6.3 | 1.00 | 0 | 0.368 |
| No | 29.3 | | 20.2 | | 13.4 | | 7.6 | | 5.3 | |
| **Itchy nose or throat** | | | | | | | | | | |
| Yes | 32.2 | 0.473 | 20.3 | 1.00 | 6.8 | 0.249 | 8.5 | 0.872 | 5.1 | 1.00 |
| No | 26.5 | | 20.1 | | 13.3 | | 6.8 | | 5.2 | |
| **Muscle pain** | | | | | | | | | | |
| Yes | 20.7 | 0.062 | 21.6 | 0.716 | 0.9 | **<0.001** | 6.3 | 0.853 | 2.7 | 0.229 |
| No | 31.3 | | 19.2 | | 18.2 | | 7.6 | | 6.6 | |
| **Runny nose** | | | | | | | | | | |
| Yes | **33.5** | **0.011** | 19.9 | 1.00 | 9.1 | 0.100 | 8.0 | 0.678 | **8.0** | **0.024** |
| No | **19.7** | | 20.4 | | 15.9 | | 6.1 | | **1.5** | |
| **Wheezing** | | | | | | | | | | |
| Yes | 32.4 | 0.616 | 14.7 | 0.542 | 20.5 | 0.177 | 14.7 | 0.144 | 8.8 | 0.548 |
| No | 26.6 | | 20.8 | | 10.9 | | 6.2 | | 4.7 | |
| **Chest pain** | | | | | | | | | | |
| Yes | **15.3** | **0.014** | 25.0 | 0.313 | 9.7 | 0.634 | 2.8 | 0.167 | 4.2 | 0.884 |
| No | **30.9** | | 18.6 | | 12.7 | | 8.5 | | 5.5 | |
| **Asthma** | | | | | | | | | | |
| Yes | 25.0 | 0.945 | 12.5 | 0.476 | 8.3 | 0.836 | 8.3 | 1.00 | **16.7** | **0.031** |
| No | 27.9 | | 20.8 | | 12.0 | | 7.1 | | **4.2** | |
| **Hypertension** | | | | | | | | | | |
| Yes | 25.0 | 0.945 | 20.8 | 1.00 | 4.2 | 0.385 | 0.0 | 0.315 | 0.0 | 0.473 |
| No | 27.9 | | 20.1 | | 12.4 | | 7.7 | | 5.6 | |
| **Sinusitis** | | | | | | | | | | |
| Yes | 13.8 | 0.124 | 6.9 | 0.103 | 3.4 | 0.249 | 6.9 | 1.00 | 0.0 | 0.375 |
| No | 29.1 | | 21.6 | | 12.6 | | 7.2 | | 5.8 | |
| **Sore throat** | | | | | | | | | | |
| Yes | 23.6 | 0.178 | 19.3 | 0.825 | 11.4 | 0.896 | 10.0 | 0.075 | 4.3 | 0.681 |
| No | 31.1 | | 21.0 | | 12.6 | | 4.2 | | 6.0 | |

*Determinants of infection were predicted using univariate analysis. Bold values indicate statistically significant results. Bold values indicate statistically significant results.

panel did not include other well-known bacterial and viral respiratory pathogens, such as *Streptococcus pneumoniae*, *Haemophilus influenzae*, *Mycobacterium tuberculosis*, *Klebsiella pneumoniae*, *Acinetobacter baumannii*, *Legionella pneumophila*, measles virus, human bocavirus, enterovirus D68, herpesviruses, and parvovirus B19. For instance, the

**Table 4. Determinants of pathogens among patients suffering from acute community-acquired upper respiratory infections using multivariable logistic regression models in Lebanon.**

| | Human rhinovirus/ enterovirus | | Influenza A virus | | Respiratory syncytial virus | | SARS-CoV-2 | | Parainfluenza virus | |
|---|---|---|---|---|---|---|---|---|---|---|
| | Backward multivariable logistic regression model | | | | | | | | | |
| | adj. OR (IC95%) | P-value | adj. OR (IC95%) | P-value | adj. OR (IC95%) | P-value | adj. OR (IC95%) | P-value | adj. OR (IC95%) | P-value |
| **Age** | | | | | | | | | | |
| ≤5 years[1] | | | | | | | | | | |
| 6–17 years | | | 3.52 (0.99-13.2) | 0.054 | | | | | | |
| ≥18 years | | | 2.95 (0.77-1.21) | 0.122 | | | | | | |
| **Region** | | | | | | | | | | |
| Rural | | | | | | | | | | |
| Urban[1] | | | | | | | | | | |
| **Educational level** | | | | | | | | | | |
| High school or above | | | **0.23 (0.06-0.76)** | **0.019** | **0.29 (0.10-0.76)** | **0.017** | 3.55 (1.09-14.7) | 0.051 | | |
| Less than High School[1] | | | | | | | | | | |
| **Marital status** | | | | | | | | | | |
| Single | | | | | | | | | 4.06 (0.70-77.2) | 0.196 |
| Married[1] | | | | | | | | | | |
| **Vaccinated against influenza** | | | | | | | | | | |
| Yes | | | 0.00 (0-2.2e$^{20}$) | 0.993 | | | | | | |
| No[1] | | | | | | | | | | |
| **Citizenship** | | | | | | | | | | |
| Syrian or Palestinian refugee | | | 4.24 (0.81-24.4) | 0.090 | | | | | **7.22 (1.19-37.0)** | **0.020** |
| Lebanese[1] | | | | | | | | | | |
| **Season** | | | | | | | | | | |
| Other seasons | | | 0.00 (0-8.0e$^{17}$) | 0.990 | 0.00 (0-6.1e$^{28}$) | 0.991 | | | | |
| Fall[1] | | | | | | | | | | |
| **Cough** | | | | | | | | | | |
| Yes | | | **3.33 (1.18-11.2)** | **0.034** | | | | | | |
| No[1] | | | | | | | | | | |
| **Fever** | | | | | | | | | | |
| Yes | 0.57 (0.31-1.05) | 0.072 | **7.63 (3.19-20.9)** | **<0.001** | | | | | | |
| No[1] | | | | | | | | | | |
| **Nausea** | | | | | | | | | | |
| Yes | | | | | | | | | | |
| No[1] | | | | | | | | | | |
| **Vomiting** | | | | | | | | | | |
| Yes | | | **4.06 (1.39-12.4)** | **0.011** | | | | | | |
| No[1] | | | | | | | | | | |
| **Dyspnea** | | | | | | | | | | |
| Yes | | | | | **2.69 (1.11-6.66)** | **0.029** | | | | |
| No[1] | | | | | | | | | | |

*(Continued)*

**Table 4.** (Continued)

| | Human rhinovirus/ enterovirus | | Influenza A virus | | Respiratory syncytial virus | | SARS-CoV-2 | | Parainfluenza virus | |
|---|---|---|---|---|---|---|---|---|---|---|
| | Backward multivariable logistic regression model | | | | | | | | | |
| | adj. OR (IC95%) | P-value | adj. OR (IC95%) | P-value | adj. OR (IC95%) | P-value | adj. OR (IC95%) | P-value | adj. OR (IC95%) | P-value |
| **Headache** | | | | | | | | | | |
| Yes | | | | | | | | | | |
| No[1] | | | | | | | | | | |
| **Malaise** | | | | | | | | | | |
| Yes | | | | | | | | | | |
| No[1] | | | | | | | | | | |
| **Moderate-to-severe diarrhea** | | | | | | | | | | |
| Yes | | | | | | | | | | |
| No[1] | | | | | | | | | | |
| **Muscle pain** | | | | | | | | | | |
| Yes | | | | | **0.08 (0.0-0.42)** | **0.017** | | | | |
| No[1] | | | | | | | | | | |
| **Runny nose** | | | | | | | | | | |
| Yes | **4.16 (2.19-8.28)** | **<0.001** | | | **0.37 (0.15-0.90)** | **0.031** | | | 7.05 (1.26-132) | 0.069 |
| No[1] | | | | | | | | | | |
| **Wheezing** | | | | | | | | | | |
| Yes | | | | | | | **6.22 (1.58-24.8)** | **0.008** | | |
| No[1] | | | | | | | | | | |
| **Chest pain** | | | | | | | | | | |
| Yes | **0.43 (0.19-0.91)** | **0.034** | | | | | **0.12 (0.02-0.56)** | **0.018** | | |
| No[1] | | | | | | | | | | |
| **Asthma** | | | | | | | | | | |
| Yes | | | | | | | | | 3.94 (0.66-18.6) | 0.099 |
| No[1] | | | | | | | | | | |
| **Sinusitis** | | | | | | | | | | |
| Yes | 0.43 (0.11-1.28) | 0.157 | | | | | | | | |
| No[1] | | | | | | | | | | |
| **Sore throat** | | | | | | | | | | |
| Yes | | | | | | | 2.68 (0.93-8.49) | 0.076 | | |
| No[1] | | | | | | | | | | |

Variables with a P-value ≤ 0.20 in the univariate analysis were included in Model 1 (multivariable logistic regression analysis, Supplementary S2 Table). To further refine the model, a backward logistic regression model was performed including only complete cases. [1]Reference group. Bold values indicate statistically significant results.

**Table 5. Association between common viruses (Prevalence > 4%) among patients suffering from acute community-acquired upper respiratory infections using multivariable logistic regression models in Lebanon.**

| | Backward Multivariable Logistic Regression Model | | |
|---|---|---|---|
| | **adj. OR** | **95% CI** | **P-value** |
| **Human rhinovirus/enterovirus** | | | |
| Influenza A virus | **0.25** | **0.10-0.54** | **0.001** |
| Respiratory syncytial virus | | | |
| SARS-CoV-2 | **0.21** | **0.03-0.75** | **0.039** |
| Parainfluenza virus | | | |
| **Influenza A virus** | | | |
| Human rhinovirus/enterovirus | **0.24** | **0.10-0.53** | **0.001** |
| **Respiratory syncytial virus** | 0.40 | 0.11-1.06 | 0.096 |
| **SARS-CoV-2** | 0.13 | 0.01-0.67 | 0.053 |
| **Parainfluenza virus** | | | |
| **Respiratory syncytial virus** | | | |
| **Human rhinovirus/enterovirus** | | | |
| **Influenza A virus** | 0.46 | 0.13-1.21 | 0.156 |
| **SARS-CoV-2** | | | |
| **Parainfluenza virus** | | | |
| **SARS-CoV-2** | | | |
| **Human rhinovirus/enterovirus** | **0.20** | **0.03-0.71** | **0.033** |
| **Influenza A virus** | 0.13 | 0.01-0.66 | 0.052 |
| **Respiratory syncytial virus** | 0.28 | 0.02-1.44 | 0.226 |
| **Parainfluenza virus** | 0.00 | $0-9e^{24}$ | 0.992 |
| **Parainfluenza virus** | | | |
| **Human rhinovirus/enterovirus** | | | |
| **Influenza A virus** | | | |
| **Respiratory syncytial virus** | | | |
| **SARS-CoV-2** | 0.00 | $0-2.4e^{32}$ | 0.991 |

In the initial multivariable logistic regression analysis, selected respiratory tract pathogens with a prevalence > 4% were entered in the model as explanatory variables (Supplementary S3 Table). To further refine the model, a backward logistic regression model was performed including only complete cases. Bold values indicate statistically significant results.

Lebanese study communicating a prevalence of 84% of community-acquired respiratory tract infections in patients admitted to the emergency departments analyzed diverse samples of the upper and lower respiratory tracts and the presence of common bacterial pathogens, such as *S. pneumoniae*, elevating thus the prevalence compared to our study focusing on upper respiratory tract samples and more on viral pathogens [26].

Most samples were collected in the fall and winter. This was related to the number of symptomatic individuals during these seasons. Although a non-significant association by our regression models was established between a specific season and the five common pathogens or acute respiratory infections, our increased isolation of viruses (human rhinovirus/enterovirus, RSV, and influenza A virus) in the fall season and December has also been described previously [24,29]. Concerning the determinants of acute respiratory disease, people with a high education level had a significantly lower probability of infection – specifically four times less – compared to those with a low educational level; re-highlighting the previously documented protective role of a high education level in reducing the risk of acute community-acquired respiratory infections (Table 2, S1 Table) [30]. Among the listed symptoms, runny nose, and wheezing were significantly reported in infected patients than in non-infected patients with four and eight-time increases in the odds, in contrast to chest pain

and sinusitis found less in infected patients with three and five-time decreases in the odds (Table 2, S1 Table); this may be due to the non-severity and mildness of infections seen among participants. Moreover, the associations of clinical presentations with acute respiratory infections differ worldwide between studies. While nasal obstruction significantly increased the risk of acute respiratory infections among children under five years at a hospital in Praia, Santiago Island, Cabo Verde in 2019, chest pain as our study decreased the odds of acute respiratory tract infections, revealing that chest pain could be associated with other etiologies, apart from acute respiratory infections [31].

Although we have screened essentially here viruses, the dominance of viral etiology over bacterial has been evidenced in many studies [3]. The co-infection rate among positive tests was 16.5% lower than that previously reported in Lebanon (42.9% [25]; 37% [24]) but similar to other worldwide studies using generally similar approaches during the COVID-19 pandemic [3]. This lower rate may be due to the diversity of our sample in terms of age groups, in contrast to the two Lebanese studies that focused on children and the different panels of screened pathogens. For example, our study did not screen the pathogens frequently identified with other viruses in Lebanese studies, such as human bocavirus [24]. Interestingly, we found a negative association between human rhinovirus/enterovirus, influenza A, and SARS-CoV-2 (Table 5, S3 Table) when the infection by human rhinovirus/enterovirus pathogen seems to inhibit the disease by the two others. Indeed, co-infections are complex relationships between pathogens, and *in vitro* studies have shown that a rhinovirus infection reduces the likelihood of SARS-CoV-2 infection [32].

Human rhinovirus/enterovirus (27.2%) was the most commonly detected virus in Tripoli between May 2023 and February 2024, which mirrored the figure before the COVID-19 pandemic in Lebanon of the dominance of human rhinovirus in respiratory samples [25,26]. Similarly, rhinoviruses prevailed in many studies, even during the COVID-19 pandemic [3,22,33]. Thus, the COVID-19 pandemic does not seem to have altered potentially the most common pathogen in Lebanon. As expected from its name, rhinoviruses, the common cause of colds, were more observed here in patients with a runny nose (four-fold significant increases of odds) and less with chest pain (two-fold significant decreases of odds) (Table 4, S2 Table). The same association was also observed in a Lebanese study between rhinorrhea and human rhinovirus but was insignificant [25].

Before COVID-19, influenza A had a substantial burden in Lebanon, with a hospitalization rate of 48.1 per 100,000 and several influenza A-associated hospital admissions of 2,866 [34]. After four years of COVID-19, influenza A was the second most prevalent pathogen, contributing to 19.8% of community-acquired acute respiratory tract infections in our study with influenza A virus A/H1-2009 (16.4%) being the major subtype, followed by influenza A virus A/H3 (1.9%). Influenza B infected only 0.9% of patients. The prevalence of influenza A was higher than that retrieved before COVID-19 among patients admitted to the emergency departments, with community-acquired respiratory tract infections (16%) between 2017−2018 [26], and the average annual positivity rate (14%) of laboratory-confirmed influenza at a tertiary care center in Beirut between 2008 and 2016 [35]. Such a potential increase in influenza activity after the COVID-19 pandemic has been witnessed in many countries. Indeed, following the implementation of non-pharmaceutical interventions and behavioral changes during the COVID-19 pandemic to halt the spread of SARS-CoV-2, influenza activity plummeted substantially and generally remained globally low from the early spring of 2020 to late 2021 [36]. In the WHO European region, the circulation of influenza viruses has continued to be impacted by the public health emergency measures in 2021/22, but less apparent than during the 2020/21 and 2019/20 seasons [37]. However, after the relaxation of control measures, an upswing in influenza activity was reported in many countries, but to different degrees [23,38,39]. In Canada, influenza or RSV virus were not identified among patients hospitalized with acute respiratory infections in 2020−2021; however, their detection levels increased in the following two years but remained below the levels seen before the COVID-19 pandemic [22]. In agreement with our observations, after two years of decline in Egypt, influenza and RSV detection rates outreached pre-pandemic phase rates among children under 16 years old with acute respiratory infections in Egypt outpatient clinics [40]. Using the same kit of PCR multiplex of our study, influenza A exhibited a nearly similar rate (20%) in acute respiratory tract infections at a tertiary referral pediatric center in Italy between 2022 and 2023 [23]. Such a potential

resurgence in influenza rate after COVID-19 seen in Lebanon could be partly due to the low vaccination rate (6.8%) among participants – a rate fitting well with that communicated in Lebanese studies (6%−22%) [24,35]. Other factors that could explain such an uptick are the continuous evolution of influenza viruses to evade pre-existing immunity, the accumulation of susceptible populations due to limited exposure to natural infection, or the subsequent decrease of influenza antibodies with increased human-to-human contact [38].

Influenza virus (sub)types circulate differently from season to season [41]. Usually, both types (A and B) circulated in Lebanon in the last decade. According to influenza seasons, some studies showed a predominance of type A [24,26,35], while others had a codominance or predominance of B [42,43]. Regarding the subtypes, after one year of the 2009 pandemic of H1N1, the incriminated strain (Influenza A/H1N1pdm09) along with the B influenza virus were found to co-circulate with equal prevalence in Lebanon during the 2010–2011 season, while the H3N2 virus predominated during the 2011–2012 season [43]. Influenza A/H3N2 and A/H1N1pdm09 have spread at high levels in Lebanon over the past decade, both before and after the COVID-19 pandemic [44]. During the 2021/2022 season, influenza A/H3 was the predominated variant in the WHO European region, co-circulating with A/H1pdm09 and B/Victoria viruses in some countries [37]. In China, influenza B/Victoria was a major subtype in December 2021 but not in 2022, with influenza H3N2 being the principal one [38]. In Turkey, while the dominant influenza subtype was influenza A/H3 in 2022, it was H1 in 2023 [19]. In Lebanon, the distribution of subtypes is likened to the situation in Turkey, with the dominance of H3 in 2021–2022 but influenza A, particularly A/H1N1pdm09 in 2023 [45]. In our study done from May 2023 to February 2024, influenza virus A/H3 (1.9%) constituted only a fraction of influenza A subtypes, with influenza A/H1N1pdm09 being the major subtype, corroborating thus with Lebanese data [45]. It is worth mentioning that Influenza B (Victoria lineage) dominated in February 2024 according to Lebanese data [45], in contrast to our findings, possibly due to the low number of tested samples during this month. Interestingly, a fraction of patients (1.5%) were infected with influenza A subtypes not belonging to H1 and H3 and perhaps to H5, H7, and H9, among others [41].

Regarding influenza symptoms, fever and cough have been identified as potential predictors of infection [46], increasing the odds of influenza by eight- and three-fold, respectively (Table 4, S2 Table). Vomiting was the only gastrointestinal symptom potentially predicting influenza infection by multiplying the odds of disease by four (Table 4, S2 Table). Similarly, a study has shown that influenza, human metapneumovirus, and enterovirus D68 were significantly more observed in patients suffering from gastrointestinal symptoms [47]. Moreover, a high education level significantly decreased the odds of influenza infection by four (Table 4, S2 Table). Such protective association was also evidenced elsewhere [48].

Our results showed that RSV is the third dominant agent detected in different age categories (11.4% of participants), with a higher percentage among children. This prevalence was higher than that reported in pre-COVID-19 studies among patients with acute respiratory tract infections presenting to the emergency department: RSV was identified in only 4% of the cases and exclusively detected in pediatric patients between 2017 and 2018 [26] and in 8% of infants and children less than 18 years old between 2009−2012 [25]. However, RSV reached potentially higher prevalence in severe cases in Lebanon; it was responsible for 26.7% of respiratory infections seen in hospitalized children whose ages ranged between 15 days and 6 years in the 2008 winter [49], and it was the second most frequent pathogen infecting 19% of the hospitalized children cases between 2013 and 2014 [24], it was the most common respiratory pathogen among pediatric cancer patients with acute respiratory tract infection between 2014−2015 [27]. Indeed, RSV is acknowledged widely as the leading cause of hospital admissions and a critical contributor to child mortality worldwide [23]. This study also found that illness caused by RSV could be more severe, as indicated by the significant probability of patients suffering from dyspnea being RSV-infected (Table 4, S2 Table) – an association also evidenced by other studies [40,49,50]. Additionally, RSV was significantly less detected herein in patients with muscle pain (12.5-fold lower odds) or runny nose (3-fold lower odds) (Table 4, S2 Table). The COVID-19 pandemic has also impacted RSV [51], but such an impact could not be assessed here accurately due to the lack of data during the COVID-19 pandemic. In some countries, RSV has emerged as a primary pathogen responsible for acute respiratory tract infections and showed even sometimes atypical activity by

preceding rhinoviruses [19,23,52]. In Turkey, RSV ranked after rhinoviruses among cases with suspected respiratory tract infections between 2021 and 2023 and shared a similar detection rate (10.57%) with our study [19]. In Egypt, RSV was listed subsequent influenza and was liable for 20.9% of cases during an outpatient clinic survey of infants and children in October 2022 [40]. Finally, RSV was also encountered less in patients with high education levels, having three times lower odds of infection than patients with low education levels (Table 4, S2 Table), as observed elsewhere [53].

SARS-CoV-2 was the fourth most prevalent pathogen, infecting 6.8% of patients with acute respiratory symptoms. This infection percentage was in line with the average PCR positivity rate (~6.3%) of the Lebanese population during this period (between May 2023 and January 2024, excluding August) calculated from the available data from the Lebanese Ministry of Public Health. Our percentage indicates that SARS-CoV-2 continued to circulate among the Lebanese population during the study period as one of the essential pathogens but at a prevalence lower than recorded during previous COVID-19 waves in Lebanon (~21% during January and February 2022 of wave 4) and higher than reported at the beginning of the pandemic during effective measures (1.2% during April and May 2020) [9]. A similar decreasing trend was also observed among pediatric outpatients with acute respiratory tract infections in the United States (positivity rates of 9.1% of tests in 2020, 11.4% in 2022, and 3.6% in 2023) [3]. This relatively lower COVID-19 infection percentage compared to previous rates may be due to the developed immunity in the Lebanese residents either through vaccination or previous infections. However, among the 299 patients aware of their vaccination status, only seven (2.3%) reported receiving a COVID-19 vaccine in the past six months. Unfortunately, we did not have information on their exact COVID-19 vaccination status, past COVID-19 infections, or serum antibody levels in these patients, hindering our understanding of such a prevalence. Meanwhile, a recent study in Lebanon revealed that most vaccinated participants with two doses of SARS-CoV-2 vaccines had a high immune response (>249 U/ml) after 6 months of the second dose [54]. Although October 2023 recorded the highest detection of SARS-CoV-2 during the study period, surveillance data for COVID-19 from this month were absent from the Lebanese Ministry of Public Health, and no cases were reported to the WHO [8]. This highlights the critical need for robust case tracking of COVID-19 in Lebanon. Intriguingly, despite being insignificant, 9.8% of adults were infected by SARS-CoV-2 compared to only 2.2% of children aged between 6–18 years old and 4.2% of preschool-aged children. Similar variations in pathogen distribution have been observed elsewhere. In Japan, SARS-CoV-2 was the leading pathogen among older outpatients with respiratory symptoms during the COVID-19 pandemic, in contrast to the pediatric population [21]. Comparable trends were also reported in Cameroon and Turkey [19,55]. However, these discrepancies did not imply a lower SARS-CoV-2 infection rate in children but maybe lower symptomatology and an underdiagnosis because many cases of children are generally asymptomatic or mild [56]; since this study did not examine asymptomatic children but only those with acute respiratory tract symptoms thus dropping the prevalence of COVID-19 infection in this category. Similarly, a Lebanese retrospective study revealed a marked decline in the proportion of school-age children (≥3 years old) hospitalized with respiratory tract infections during the COVID-19 era compared to the pre-COVID-19 era despite the significant increase in the overall percentage of Lebanese hospitalized children across different age groups during the COVID-19 era [57]. In Quebec, Canada, respiratory viruses other than SARS-CoV-2 were the major contributor to pediatric acute respiratory tract infections' hospitalization during all three first pandemic years. SARS-CoV-2 was absent among children with acute respiratory tract hospitalizations during the first pandemic year and infrequently observed during the second and third years [22]. In adults, SARS-CoV-2 was the major contributor to acute respiratory tract infections' hospitalization during the first two pandemic years, but its relative importance gradually lessened during the third pandemic year [22]. Besides, our data showed that asthma is not associated with an elevated risk of contracting COVID-19 infection, as earlier shown [58]. However, the likelihood of SARS-CoV-2 infection was significantly higher in patients with wheezing by six times and lower in patients with chest pain by eight times (Table 4, S2 Table). Interestingly, COVID-19 symptoms may differ according to the circulating variant. In Saudi Arabia, symptoms like shortness of breath, wheezes, myalgia, tachypnea, and respiratory distress were significantly more frequent in the second wave than in the first wave [59].

Four types of PIV with antigenic, genetic, and even clinical differences have been identified. PIV1 and PIV2 frequently cause croup and cold-like symptoms and PIV3 bronchiolitis and pneumonia [60,61]. PIV4 is less well characterized and appears most commonly associated with mild and asymptomatic diseases with upper respiratory tract symptoms but could also lead to severe disease [61–63]. PIV3 is the prevalent type frequently detected worldwide [60,63]. PIV4 is less prevalent than other types, possibly due to its limited testing and omission from many respiratory virus diagnostic panels [60]. In our study, when amalgamating all types, PIV infected 5.2% of patients, with PIV4 (2.5%) being the most common, followed by PIV3 (1.5%), PIV1 (1.2%), and PIV2 (0.3%). This prevalence resembled the positivity rate for any PIV type (5%) between 2011 and 2019 in the USA, with PIV3 being the frequent type (55%), followed by PIV1 (18%), PIV2 (14%), and PIV4 (13%) [60]. The prevalence of PIV here (5.2%) appeared to be lower than that reported in pre-COVID-19 studies conducted in Lebanon [25,26]. According to the literature, the COVID-19 pandemic has also impacted PIV [21,23,52,64]. Our prevalence was also consistent with the study done in Italy (5.4%, 2022/2023) [23]. Studies conducted in Lebanon have listed different frequency rates of PIV types, even though PIV3 was one of the most prevalent types in Lebanon [24–27]. PIV4 was the first dominant type among hospitalized children in 2013−2014 [24], the second type among children presenting to emergency departments in 2009−2012 [25], a rare type among patients presenting to emergency departments in 2017−2018 [26], and absent among patients with pediatric cancer patients suffering from febrile episodes with upper respiratory tract symptoms [27]. So, other studies must be done in Lebanon to assess its precise prevalence, trends, and clinical presentation. PIV3 and PIV4 were also dominant in China among hospitalized children with acute lower respiratory infections [63]. Noteworthy, the trends of PIV3 in post-COVID-19 varied by country, showing either a surge or a decline [21,52,65]. Interestingly, we found a significant association between Syrian and Palestinian refugees and PIV, with a seven-fold increase in odds compared to the host Lebanese population. However, no data regarding the epidemiology of PIV were unfortunately available from Syrians and Palestinians to delve further. In addition, the low number of Syrians and Palestinian refugees in our sample and the absence of genomic analysis prevent us from proving a local outbreak.

Classical seasonal human coronaviruses (NL63, 229E, OC43, HKU1) represented together 6.1% of patients with acute respiratory tract infections, a similar prevalence to that communicated in other Lebanese studies (~6%) [24,26]. Among these coronaviruses, HCoV-NL63 (2.8%) was the most common, pursued by HCoV-229E and HCoV-OC43 (each 1.5%), and HCoV-HKU1 (0.3%). The epidemiology of human coronaviruses was depicted poorly in Lebanon, with only one study characterizing them and demonstrating a preponderance of HCoV-OC43 [27]. HCoV-OC43 was generally the prevalent human coronavirus [19,21,23,66]. The conclusion of a potential increased prevalence of NL63 compared to other Lebanese studies may be tricky because HCoV-NL63 is underestimated in Lebanon. Although HCoV-NL63 shares the same host receptor (ACE2) as SARS-CoV-2, it typically causes mild, self-limiting upper respiratory tract infections and, occasionally, more severe lower respiratory tract infections [67,68]. A recent study in Malawi revealed a frequent detection of HCoV-NL63 in patients with influenza-like illness and HCoV-229E in patients with severe acute respiratory disease [69], explaining thus possibly the high recovery of HCoV-NL63 compared to other seasonal coronaviruses in our outpatients with mild acute respiratory tract infections. Another scenario could be a different circulation of coronavirus strains. A rare localized epidemic of HCoV-NL63 among pediatric patients with respiratory diseases in Guilin, China, in 2021−2022 was explained by the emergence of a new coronavirus HCoV-NL63 genotype [67]. A recent study also described an alternating pattern of circulation between alpha-coronaviruses (229E and NL63) and beta-coronaviruses (HKU1 and OC-43), with one alpha-coronavirus dominating in one season and a beta-coronavirus dominating in the following season [70].

The prevalence of adenovirus (1.9%) in this study was lower than in previous Lebanese studies (7–10%) [24,26], and studies done in post-COVID-19 in Japan, Turkey, and Italy [19,21,23], but higher than in another Lebanese study (0%) [25]. Previous data indicate that the COVID-19 pandemic had an inconsistent effect on adenovirus rates, with a decline in rates from 2019 to 2021 in China [52], while rates remained unchanged in Saudi Arabia during the pre- and post-COVID-19 periods [71].

Intriguingly, metapneumovirus was not identified among participants, although its prevalence has reached 16% among Lebanese patients presented at emergency departments in a previous study [26]. One reason for this low prevalence could be the type of investigated samples (upper vs. lower respiratory tract), as the previously cited Lebanese study tested specimens of the upper and lower respiratory tracts [26]. The limited rates were also retrieved in post-COVID-19 periods in other countries [19,21,23].

Concerning bacteria, we detected four cases of *B. pertussis*; three were aged 5 years or younger (6 weeks, 4, and 5 years), and one was 40. Pertussis, commonly known as whooping cough, is a vaccine-preventable respiratory tract infectious disease and a notifiable communicable disease in Lebanon. In 2023, about 101 pertussis cases were reported, according to WHO, with an incidence of 17.5 per 1,000,000 total population [72]. The reported count of cases varied between 0 and 97 from 2000 to 2019, while data were unavailable between 2020 and 2022 [72]. This detection may be attributed to the suboptimal vaccination coverage, with WHO/UNICEF estimates of national immunization coverage of 78% and 55% for the first and third dose of DTP (diphtheria, tetanus, and pertussis)-containing vaccine, respectively [72]. Unfortunately, we do not have information about the extent of DTP vaccination among participants. Detecting an infant younger than 6 months here highlights the importance of DTP vaccination during pregnancy, as infants have not yet completed their primary immunization course [73–75]. Noteworthy, some countries with sustained high coverage with acellular pertussis vaccine have also witnessed a pertussis resurgence [76], underscoring the significance of surveillance even after years of vaccine introduction.

*M. pneumoniae* was only found in 0.6% of patients, a prevalence lower than 8% reported among 100 patients presenting in emergency departments in Lebanon [26]. Aligned with our study, *M. pneumoniae* was not or rarely identified in Italy, Turkey, and Japan [19,21,23]. Interestingly, we have not detected *B. parapertussis* and *C. pneumoniae* in our study, even the reports depicting increased detection rates in many regions after COVID-19 [20,77,78], requiring thus real-time tracking of their accurate trends in Lebanon.

Finally, the authors acknowledge some limitations. First, although the assay adopted in this study (BioFire® FilmArray® Respiratory Panel 2.1 Plus) screens the major viral pathogens involved in respiratory tract infections, it is not exhaustive. The assay does not differentiate between human rhinovirus and enterovirus, nor test enterovirus D68, a significant enterovirus type [23]. Our data is limited to North Lebanon and cannot be generalized to the entire country. The study focused only on the prevalence of infections caused by specific viruses among symptomatic patients, so the actual circulation of these viruses could be higher, as we did not screen asymptomatic individuals. In the discussion, we compared our findings with pre-COVID-19 Lebanese studies and speculated on potential increases or decreases in specific viral infections. However, this comparison may be misleading, as these studies were conducted in different spatiotemporal contexts and may vary in study populations and methodologies. A longitudinal study that analyzes respiratory samples before, during, and after the onset of COVID-19 is urgently needed to better understand the impact of COVID-19 on the landscape of respiratory infections.

## Conclusion

This study offers a preliminary yet valuable insight into the epidemiology of community-acquired acute respiratory infections in North Lebanon during the post-COVID-19 era (2023−2024). Our findings reveal the continued prominence of viral etiologies in respiratory diseases. After four years of the COVID-19 pandemic, SARS-CoV-2 was still circulating but ranked fourth after human rhinovirus/enterovirus, influenza A, and RSV. The notable presence of vaccine-preventable infections, such as *B. pertussis*, highlights the urgent need to strengthen vaccine coverage across the Lebanese population. The high infection rates of influenza A, SARS-CoV-2, and RSV further emphasize the necessity of boosting vaccination efforts for viruses with available vaccines and prioritizing the development of vaccines for those without, such as PIV. Ultimately, this study provides crucial data to help clinicians, health authorities, and stakeholders better understand the evolving epidemiologic landscape and develop targeted strategies for respiratory disease diagnosis, prevention, and

antimicrobial stewardship. Moreover, real-time, comprehensive surveillance utilizing advanced molecular techniques is essential for closely monitoring the circulation and trends of respiratory pathogens, thereby informing timely and effective public health responses.

## Supporting information

**S1 Table. Sociodemographic and risk factor determinants of acute community-acquired upper respiratory infections in the study population.**
(DOCX)

**S2 Table. Determinants of pathogens among patients suffering from acute community-acquired upper respiratory infections using multivariable logistic regression models in Lebanon.**
(DOCX)

**S3 Table. Association between common viruses (Prevalence > 4%) among patients suffering from acute community-acquired upper respiratory infections using multivariable logistic regression models in Lebanon.**
(DOCX)

## Acknowledgments

This work was supported by a grant from the Lebanese University, Al Hamidy Medical Charitable Center, the GABRIEL Network, and Mérieux Foundation. The funders had no role in study design, data collection and analysis, decision to publish, or preparation of the manuscript. The authors would like to thank the LMSE staff members for their assistance in this project. They would also like to thank Dr. Marcel Achkar and all the participating pediatricians for collaborating in this study: Dr. Amer Ghaouche, Dr. Amer Barakeh, Dr. Ahmad Malas, Dr. Nazih Kamaledine, Dr. Randa Jamal Akoum, and Dr. Aya Traboulsi.

## Author contributions

**Conceptualization:** Rayane Rafei, Marwan Osman, Fouad Dabboussi, Monzer Hamze.

**Formal analysis:** Rayane Rafei, Marwan Osman, Fouad Dabboussi, Monzer Hamze.

**Funding acquisition:** Monzer Hamze.

**Investigation:** Rayane Rafei, Marwan Osman, Bashir Amer Barake, Hassan Mallat, Fouad Dabboussi, Monzer Hamze.

**Validation:** Rayane Rafei, Marwan Osman, Bashir Amer Barake, Hassan Mallat, Fouad Dabboussi, Monzer Hamze.

**Visualization:** Rayane Rafei, Marwan Osman.

**Writing – original draft:** Rayane Rafei, Marwan Osman.

**Writing – review & editing:** Bashir Amer Barake, Hassan Mallat, Fouad Dabboussi, Monzer Hamze.

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
