## [Decision Letter · Decision Letter 0]

19 May 2025

Dear Dr. Osman,

Thank you for submitting your manuscript to PLOS ONE. After careful consideration, we feel that it has merit but does not fully meet PLOS ONE’s publication criteria as it currently stands. Therefore, we invite you to submit a revised version of the manuscript that addresses punctually all the points raised during the review process.

We look forward to receiving your revised manuscript.

Kind regards,

Flora De Conto, Ph.D.

Academic Editor

PLOS ONE

[This work was supported by a grant from the Lebanese University, Al Hamidy Medical Charitable Center, the GABRIEL Network, and Mérieux Foundation. The funders had no role in study design, data collection and analysis, decision to publish, or preparation of the manuscript.]

 [This work was supported by a grant from the Lebanese University, Al Hamidy Medical Charitable Center, the GABRIEL Network, and Mérieux Foundation. The funders had no role in study design, data collection and analysis, decision to publish, or preparation of the manuscript.]

Reviewers' comments:

Reviewer's Responses to Questions

**Comments to the Author**

1. Is the manuscript technically sound, and do the data support the conclusions?

Reviewer #1: Yes

Reviewer #2: Yes

2. Has the statistical analysis been performed appropriately and rigorously?

Reviewer #1: Yes

Reviewer #2: Yes

3. Have the authors made all data underlying the findings in their manuscript fully available?

Reviewer #1: Yes

Reviewer #2: Yes

4. Is the manuscript presented in an intelligible fashion and written in standard English?

Reviewer #1: Yes

Reviewer #2: Yes

Reviewer #1: General comments

The authors presented a study based in Tripoli, North Lebanon, screening adults and children with acute respiratory tract symptoms for a range of pathogens using a multiplex assay. Of particular focus was the observation of any changes in the predominance of pathogens before the COVID-19 pandemic and after (2023-2024). Extensive data and analyses were presented, however several details in the methodology require clarification.

Introduction

Line 59-62: Suggest checks for language - The line describing the four emerging respiratory pathogens is long and perhaps should be split into two sentences. “…in the last decades…” may be better worded as “…in the past few decades…”

Line 68: “…and totaled more than 1.2 million confirmed cases…”. Do the authors mean 1.2 million confirmed cases were recorded since the first confirmed case i.e. from 21 February 2020 to 19 December 2023?

Line 71-74: Please elaborate briefly on the listed factors and how they contributed to the “four waves at the end of June 2022”. What are the four waves, and did they occur in quick succession nationwide (or only in a particular region)?

Materials and Methods

Line 112: Is “AZM” an abbreviation, or it is part of the name i.e. Azm Center for Research in Biotechnology?

The Study Design section would benefit from additional detail:

Line 119: As a multicenter study, how many sites were involved? The authors noted one Hospital and paediatric clinics in Tripoli. Were these private primary care clinics located throughout the city?

Line 120: Since any patient presenting with acute community-acquired infections were included in the study, I would suggest detailing the inclusion/exclusion criteria. Was there any limit to the age range? Were eligible subjects diagnosed clinically only, based on specific symptoms/presentations? What was the case definition, especially since only acute cases were recruited and were there any exclusion criteria e.g. specific conditions?

Line 123: What samples were collected exactly? I noted that nasopharyngeal swabs were collected, and this was only mentioned later in line 133. How were the swabs collected, e.g. using sterile flocked swabs, since both children and adults were sampled? Especially in the case of infants, the type of flexible swab used may differ and collection undertaken by a paediatrician.

Line 126: Was vaccination history also collected as part of the questionnaire? COVID-19 and influenza vaccination was mentioned in the results and discussion but it is unclear if this data was also collected.

Line 129: Are there any references pertaining to the assumptions used in the sample size calculation using the Raosoft sample size calculator?

Line 133: Which transport/storage media was used to store the swab samples, e.g. Amies, UTM? How long were samples kept at 4C prior to transportation to the laboratory, and were they stored frozen prior to testing?

Line 135: Please provide some detail on the workflow for the BioFire® FilmArray® Respiratory Panel 2.1 Plus for pathogen screening. Were samples tested directly, using a set volume of the storage media? Was nucleic acid extraction required? Did all processes take place in an automated system? As part of the Biofire® system, this screening method appears to be a type of automated multiplex PCR technology.

Results

Line 170: The sample size was noted as 246 (Line 129), and the reported enrolment was 324. What was the reason for the higher number of recruited subjects?

Figure 1: Visually the axes labels would be clearer if they were spaced out further from the graph itself.

Figure 2: At first glance, Influenza A/H1-2009 at 16.4% (and H3) appears to be a separate group, when it is actually part of the total 19.8% under influenza A. Suggest to explain this clearly in the figure captions. As the prevalence is expressed as a percentage, (%) should also be added to the y-axis e.g. “prevalence of infection (%)”.

Line 224: What do the authors mean by “..rhinovirus/enterovirus had two notable peaks with the October peak longer than December.”? Was there a higher percentage of infection in October, or for a longer period of time? What was the difference in percentage?

Table 4: I would suggest a summarised table with key findings from the logistic regression model be included in the main body of the manuscript, and the full table shared as supplementary data.

Reviewer #2: it is an interesting study but it needs minor revision.

please add new references in support, not more than 6 yrs preferably because covid 19 studies are a lot.

please arrange a graphical abstract image for your data

write brief conclusion and limitations before it

**Do you want your identity to be public for this peer review?** For information about this choice, including consent withdrawal, please see our Privacy Policy

Reviewer #1: No

Reviewer #2: **Yes: ** Hamza Islam

---

## [Author Response · Author response to Decision Letter 1]

12 Jul 2025

Editor’s comments:

Comment 1.

https://journals.plos.org/plosone/s/file?id=ba62/PLOSOne_formatting_sample_title_authors_affiliations.pdf..

REPLY: The revised manuscript meets the PLOS ONE requirement.

Comment 2. Please include a complete copy of PLOS’ questionnaire on inclusivity in global research in your revised manuscript. Our policy for research in this area aims to improve transparency in the reporting of research performed outside of researchers’ own country or community. The policy applies to researchers who have travelled to a different country to conduct research, research with Indigenous populations or their lands, and research on cultural artefacts. The questionnaire can also be requested at the journal’s discretion for any other submissions, even if these conditions are not met. Please find more information on the policy and a link to download a blank copy of the questionnaire here: https://journals.plos.org/plosone/s/best-practices-in-research-reporting. Please upload a completed version of your questionnaire as Supporting Information when you resubmit your manuscript.

REPLY: Done.

Page 8, Lines 191-195.

“Inclusivity in global research”

“Additional information regarding the ethical, cultural, and scientific considerations specific to inclusivity in global research is included in the S1 Checklist.”

Comment 3. Thank you for stating the following in the Acknowledgments Section of your manuscript:

[This work was supported by a grant from the Lebanese University, Al Hamidy Medical Charitable Center, the GABRIEL Network, and Mérieux Foundation. The funders had no role in study design, data collection and analysis, decision to publish, or preparation of the manuscript.]

[This work was supported by a grant from the Lebanese University, Al Hamidy Medical Charitable Center, the GABRIEL Network, and Mérieux Foundation. The funders had no role in study design, data collection and analysis, decision to publish, or preparation of the manuscript.]

REPLY: We have removed the funding statement section from the manuscript and would like to update the acknowledgments section as follows:

Page 36, Lines 635-642

This work was supported by a grant from the Lebanese University, Al Hamidy Medical Charitable Center, the GABRIEL Network, and Mérieux Foundation. The funders had no role in study design, data collection and analysis, decision to publish, or preparation of the manuscript. The authors would like to thank the LMSE staff members for their assistance in this project. They would also like to thank Dr. Marcel Achkar and all the participating pediatricians for collaborating in this study: Dr. Amer Ghaouche, Dr. Amer Barakeh, Dr. Ahmad Malas, Dr. Nazih Kamaledine, Dr. Randa Jamal Akoum, and Dr. Aya Traboulsi.

Reviewers’ comments:

Reviewer #1:

Comment 1. The authors presented a study based in Tripoli, North Lebanon, screening adults and children with acute respiratory tract symptoms for a range of pathogens using a multiplex assay. Of particular focus was the observation of any changes in the predominance of pathogens before the COVID-19 pandemic and after (2023-2024). Extensive data and analyses were presented, however several details in the methodology require clarification..

REPLY: We thank you for your thoughtful reading and for providing encouraging feedback.

Comment 2. Line 59-62: Suggest checks for language - The line describing the four emerging respiratory pathogens is long and perhaps should be split into two sentences. “…in the last decades…” may be better worded as “…in the past few decades…”.

REPLY: We split the text into two separate sentences as follows (Lines 58–63 in the untracked version).

Page 3, Lines 59-65.

Respiratory viruses have gained particular interest in the past few decades due to the concerning emergence of at least four respiratory pathogens in humans. These pathogens exhibit generally high infectivity and transmissibility overwhelming an immunologically naive population and thus threatening public health: severe acute respiratory syndrome coronavirus (SARS-CoV), influenza A H1N1 pdm09 (2009 H1N1), Middle East respiratory syndrome coronavirus (MERS-CoV), and severe acute respiratory syndrome coronavirus 2 (SARS-CoV-2) [5–7].

Comment 3. Line 68: “…and totaled more than 1.2 million confirmed cases…”. Do the authors mean 1.2 million confirmed cases were recorded since the first confirmed case i.e. from 21 February 2020 to 19 December 2023?

REPLY: We have addressed the reviewer’s comment by revising the manuscript as follows:

Page 3, Lines 68-73.

“Lebanon reported its first confirmed COVID-19 case on February 21, 2020. According to the World Health Organization (WHO), as of December 19, 2023, the country has recorded over 1.2 million confirmed cases and 10,947 deaths attributed to the virus [8].”

Comment 4. Please elaborate briefly on the listed factors and how they contributed to the “four waves at the end of June 2022”. What are the four waves, and did they occur in quick succession nationwide (or only in a particular region)?

REPLY: We have addressed the reviewer’s comment by revising the manuscript as follows:

Pages 3-4, Lines 75-91.

“However, the cumulative effect of numerous factors, including the severe economic crisis, the explosion of the Beirut port, the lifting and easing of implemented measures, and the appearance of new SARS-CoV-2 variants brought about four waves since the first confirmed case to the end of June 2022. These waves are characterized by a sustained increase followed by a decline in daily COVID-19 cases [9]. Indeed, the severe Lebanese economic and financial crisis has weakened the healthcare system and caused a drastic shortage of medical devices and essential medications, exacerbating the control of the COVID-19 epidemic in Lebanon. The massive Beirut blast on August 4, 2020 has further strained the already fragile healthcare system, resulting in 220 deaths, 6,500 people injured, and 300,000 displaced, impacting half of Beirut’s healthcare centers and leading, among other causes, to a substantial increase in COVID-19 cases in wave 1. The emergence of Alpha, Delta, and Omicron variants has also contributed in part to the witnessed Lebanese COVID-19 waves, namely waves 2 (February 2021-June 2021), 3 (July 2021-October 2021), and 4 (December 2021-June 2022), respectively [9,10]. Deployment of COVID-19 vaccines launched on February 14, 2021, succeeded in covering 50.3% of the Lebanese population for the first dose as of December 12, 2022, and decreased the hospitalization rates during the third and fourth waves of the pandemic compared to the second wave [9].”

Comment 5. Materials and Methods. Line 112: Is “AZM” an abbreviation, or it is part of the name i.e. Azm Center for Research in Biotechnology?

REPLY: Corrected.

Comment 6. The Study Design section would benefit from additional detail: Line 119: As a multicenter study, how many sites were involved? The authors noted one Hospital and paediatric clinics in Tripoli. Were these private primary care clinics located throughout the city?.

REPLY: We thank the reviewer for this valuable comment. In this study, we recruited patients from the community through various healthcare facilities, including Al Hamidy Medical Charitable Center, Nini Hospital, and four private primary care pediatric clinics located throughout Tripoli, the capital of the North Governorate. This was clarified in the manuscript.

Comment 7. Since any patient presenting with acute community-acquired infections were included in the study, I would suggest detailing the inclusion/exclusion criteria. Was there any limit to the age range? Were eligible subjects diagnosed clinically only, based on specific symptoms/presentations? What was the case definition, especially since only acute cases were recruited and were there any exclusion criteria e.g. specific conditions?

REPLY: We have addressed the reviewer’s comment by revising the manuscript as follows:

Page 6, Lines 136-142.

“The population eligible for the study included patients of all ages who were clinically diagnosed with acute community-acquired respiratory infections and attended one of the healthcare facilities between May 2023 and February 2024. Inpatients and patients receiving antibiotics were excluded from the study. After completing an informed consent, the patient was included in this study, and a nasopharyngeal swab was collected using a sterile flocked swab. Due to the academic recess at Lebanese University, sample collection was paused at the end of July and resumed in September”.

Comment 8. Line 123: What samples were collected exactly? I noted that nasopharyngeal swabs were collected, and this was only mentioned later in line 133. How were the swabs collected, e.g. using sterile flocked swabs, since both children and adults were sampled? Especially in the case of infants, the type of flexible swab used may differ and collection undertaken by a paediatrician.

REPLY: We addressed the reviewer’s comment by revising the manuscript accordingly. Due to logistical constraints, we used the same swab types for both children and adults. Our study population ranged in age from 1 to 94 years.

Page 6, Lines 139-141

“After completing an informed consent, the patient was included in this study, and a nasopharyngeal swab was collected using a sterile flocked swab”.

Comment 9. Was vaccination history also collected as part of the questionnaire? COVID-19 and influenza vaccination was mentioned in the results and discussion but it is unclear if this data was also collected.

REPLY: We have addressed the reviewer’s comment by revising the manuscript as follows:

Page 6, Lines 143-146

“Participants or their legal representatives filled out a questionnaire covering sociodemographic characteristics (e.g., age, sex, residence, nationality, social status, educational level), respiratory and gastrointestinal symptoms (e.g., fever, runny nose, headache, cough, wheezing, diarrhea), and their vaccination status for influenza and COVID-19.”

Comment 10. Line 129: Are there any references pertaining to the assumptions used in the sample size calculation using the Raosoft sample size calculator?

REPLY: Added.

Reference [19].

“Karabulut N, Alaçam S, Şen E, Karabey M, Yakut N. The Epidemiological Features and Pathogen Spectrum of Respiratory Tract Infections, Istanbul, Türkiye, from 2021 to 2023. Diagnostics (Basel). 2024;14: 1071. doi:10.3390/diagnostics14111071 PMID: 38893598.”

Comment 11. Line 133: Which transport/storage media was used to store the swab samples, e.g. Amies, UTM? How long were samples kept at 4C prior to transportation to the laboratory, and were they stored frozen prior to testing?

REPLY: We have addressed the reviewer’s comment by revising the manuscript as follows:

Page 6, Lines 152-156

“After collection, nasopharyngeal swabs were immersed in saline medium and immediately transported in sterile containers at 4°C to the Laboratoire Microbiologie Santé et Environnement (LMSE) in Tripoli, Lebanon, where they were promptly processed”.

Comment 12. Line 135: Please provide some detail on the workflow for the BioFire® FilmArray® Respiratory Panel 2.1 Plus for pathogen screening. Were samples tested directly, using a set volume of the storage media? Was nucleic acid extraction required? Did all processes take place in an automated system? As part of the Biofire® system, this screening method appears to be a type of automated multiplex PCR technology.

REPLY: We have addressed the reviewer’s comment by revising the manuscript as follows:

Page 6, Lines 156-160

“All samples were analyzed using the BioFire® FilmArray® Respiratory Panel 2.1 Plus (bioMérieux, France), a fully automated multiplex PCR system, following the manufacturer's instructions and using a fixed volume of the sample storage medium. No additional microbiological testing (e.g., culture) was performed.”

Comment 13. Line 170: The sample size was noted as 246 (Line 129), and the reported enrolment was 324. What was the reason for the higher number of recruited subjects?

REPLY: Thank you for your valuable comment. While the minimum required sample size was estimated as 246, we aimed to increase the study population to enhance the statistical power, account for potential data loss or exclusions, and allow for more robust subgroup analyses. This approach also helped improve the generalizability of our findings.

Comment 14. Figure 1: Visually the axes labels would be clearer if they were spaced out further from the graph itself.

REPLY: Corrected.

Comment 15. At first glance, Influenza A/H1-2009 at 16.4% (and H3) appears to be a separate group, when it is actually part of the total 19.8% under influenza A. Suggest to explain this clearly in the figure captions. As the prevalence is expressed as a percentage, (%) should also be added to the y-axis e.g. “prevalence of infection (%)”.

REPLY: A sentence was added to the caption of Figure 2 (Page 9, Lines 200–222) to clarify this point.

“The infection rates of influenza A/H1-2009 and influenza A H3 viruses are presented separately as distinct groups; however, both are included in the overall percentage (19.8%) reported for influenza A.”

Comment 16. Line 224: What do the authors mean by “..rhinovirus/enterovirus had two notable peaks with the October peak longer than December.”? Was there a higher percentage of infection in October, or for a longer period of time? What was the difference in percentage?

REPLY: Thank you for this valuable comment. We have addressed the reviewer’s comment by revising the manuscript as follows:

Page 11, Lines 251-254

“Human rhinovirus/enterovirus showed two notable infection peaks, with the October peak displaying a slightly higher percentage than the one in December.”

Comment 17. Table 4: I would suggest a summarised table with key findings from the logistic regression model be included in the main body of the manuscript, and the full table shared as supplementary data.

REPLY: We provided summaries of Tables 2, 4, and 5 in the main text, while retaining the full versions as supplementary data.

Reviewer #2:

Comment 1. It is an interesting study but it needs minor revision.

REPLY: We thank you for your thoughtful reading and for providing encouraging feedback.

Comment 2. Please add new references in support, not more than 6 yrs preferably because covid 19 studies are a lot.

REPLY: Done.

Comment 3. Please arrange a graphical abstract image for your data.

REPLY: Thank you for the suggestion. We appreciate the value of graphical abstracts; however, we respectfully prefer not to include one at this time, as we believe the current presentation of the data within the manuscript and figures effectively communicates the study findings.

Comment 4. Write brief conclusion and limitations before it.

REPLY: Thank you for this valuable comment. We have addressed the reviewer’s comment by revising the manuscript as follows:

Page 34, Lines 602-614

“Finally, the authors acknowledge some limitations. First, although the assay adopted in this study (BioFire® FilmArray® Respiratory Panel 2.1 Plus) screens the major viral pathogens involved in respiratory tract infections, it is not exhaustive. The assay does not differentiate between human rhinovirus

---

## [Editor Report · Decision Letter 1]

17 Jul 2025

Shifting Respiratory Pathogens: Post-COVID-19 Trends in Community-Acquired Infections in Underserved Communities

PONE-D-25-11478R1

Dear Dr. Osman,

We’re pleased to inform you that your manuscript has been judged scientifically suitable for publication and will be formally accepted for publication once it meets all outstanding technical requirements.

Kind regards,

Flora De Conto, Ph.D.

Academic Editor

PLOS ONE
---

## [Editor Report · Acceptance letter]

PONE-D-25-11478R1

PLOS ONE

Dear Dr. Osman,

I'm pleased to inform you that your manuscript has been deemed suitable for publication in PLOS ONE. Congratulations! Your manuscript is now being handed over to our production team.

Kind regards,

on behalf of

Prof. Flora De Conto

Academic Editor

PLOS ONE